# Micromagnetic Approach to the Metastability of a Magnetite Nanoparticle and Specific Loss Power as Function of the Easy-Axis Orientation

Nathaly Roa and Johans Restrepo *

Magnetism and Simulation Group G+, Institute of Physics, University of Antioquia, A.A. 1226, Medellín 050010, Colombia; nathaly.roam@udea.edu.co
* Correspondence: johans.restrepo@udea.edu.co

**Abstract:** Magnetic nanoparticles (MNPs) have attracted a great interest in nanomedical research. MNPs exhibit many important properties. In particular, magnetic hyperthermia for selective killing of cancer cells is one of them. In hyperthermia treatment, MNPs act as nano-heaters when they are under the influence of an alternating magnetic field (AMF). In this work, micromagnetic simulations have been used to investigate the magnetization dynamics of a single-domain nanoparticle of magnetite in an external AMF. Special attention is paid to the circumstances dealing with a dynamic phase transition (DPT). Moreover, we focus on the influence of the orientation of the magnetic easy-axis of the MNP on the dynamic magnetic properties. For amplitudes of the external AMF above a certain critical value, the system is not able to follow the magnetic field and it is found in a dynamically ordered phase, whereas for larger amplitudes, the state corresponds to a dynamically disordered phase and the magnetization follows the external AMF. Our results suggest that the way the order-disorder DPT takes place and both the metastable lifetime as well as the specific loss power (SLP) are strongly dependent on the interplay between the orientation of the magnetic easy-axis and the amplitude of the external AMF.

**Keywords:** dynamic phase transition; alternating magnetic field; magnetic nanoparticle; hyperthermia; specific loss power

## 1. Introduction

It is well established that nanoparticles with sizes ranging between 1 and 100 nm exhibit different and appealing characteristics compared with their micro and macro counterparts. In recent years, magnetic nanoparticles (MNPs) have been used in areas of nanotechnology, in particular for biomedical applications [1,2] where precise delivery of anticancer drugs and selective killing of cancer cells is desirable. Magnetic hyperthermia treatment [3] is based on the use of nanoparticles dispersed in a magnetic fluid. When MNPs are under the presence of an external AMF, they are able to heat a specific area of the body, causing cell death in tumor tissue [4]. During such a process, two relaxation mechanisms for MNPs are known, namely, the Néel [5] and Brown [6] modes. In the former, magnetic moments rotate inside the nanoparticle due to the action of the external AMF. In contrast, in Brown relaxation, magnetic moments are locked to the magnetic easy-axis of the MNP and the AMF causes a mechanical rotation of the particle as a whole. The heat released to the medium is either caused by the friction between the surface of MNPs and the surrounding magnetic-fluid, or by the power dissipated through the hysteresis losses [7]. The heating efficiency of a MNP under an external AMF is expressed in the physical quantity called specific loss power (SLP) [8,9].

The dynamic response of a MNP in an external time-dependent magnetic field of amplitude $H_0$ and period $P$ is not instantaneous and it exhibits a delay in time. The time delay of the magnetization gives rise to the appearance of two possible regimes [10–12]: a

dynamically ordered phase (DOP) and a dynamically disordered one (DDP), which depend on the relationship between $P$, $H_0$ and the metastable lifetime $\tau$ [13,14]. In the DOP (see Figure 1a), magnetization is not able to follow the magnetic field, giving rise to a high value of the average magnetization. In DDP (see Figure 1b), magnetization follows the magnetic field, despite having a $P$-dependent delay, and it is able to be reversed, resulting in a well-defined hysteresis loop. In this case, the metastable lifetime, which is the elapsed time to go from the saturation state and pass through zero magnetization, can be computed.

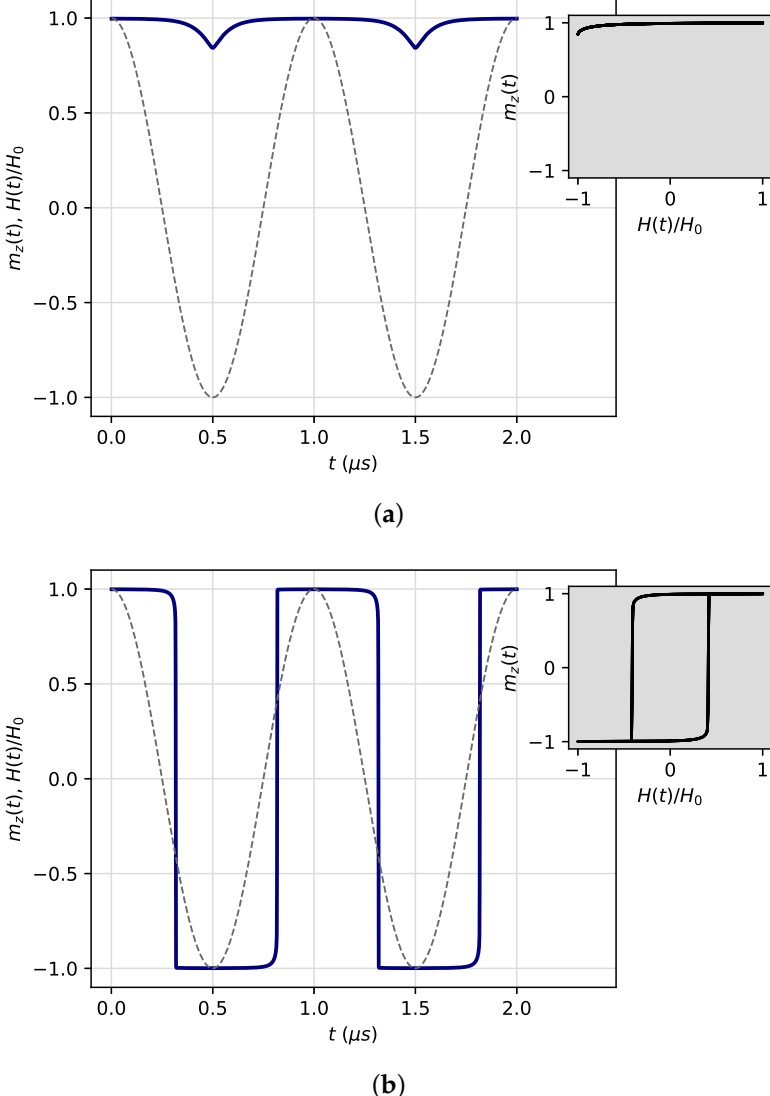

**Figure 1.** Time-dependent magnetization $m_z(t)$ (blue solid lines) of a single-domain magnetite nanoparticle of diameter 32 nm under an external AMF $H(t)/H_0$ (gray dashed lines) with amplitude (**a**) $H_0 = 80$ mT and (**b**) $H_0 = 200$ mT. The first two periods are shown for each case, where initially the sample is fully ordered magnetized. The resulting response of the system can be divided into two regimes, (**a**) a DOP one, and (**b**) a DDP one. The inset shows the magnetization $m_z(t)$ as a function of an external AMF $H(t)/H_0$ for each phase. The parameters are a frequency of $f = 1.0$ MHz, and angle of $\theta = 7.5°$ between the magnetic easy-axis of the MNP and the external AMF.

The universal aspects that allow a dynamic phase transition have been studied in some works [10–14]. B. K. Chakrabarti and M. Acharyya [12] showed that magnetic response in time by an external field depends on the competing time scales, i.e., the time period of oscillation of the external perturbation and the typical relaxation time. When the time period of oscillation is comparable with the effective relaxation time, symmetric hysteresis

loops around the origin are obtained. A breaking of the symmetry of the hysteresis appears when the driving frequency of the field increases. This phase transition depends on the field amplitude and temperature. On the other hand, G. Korniss et al. [11] studied the two-dimensional kinetic Ising model with a square-wave oscillating external field of period $P$ through Monte Carlo simulations. Their results showed that the system undergoes a dynamic phase transition when the half-period $P_{1/2}$ of the field is comparable to the metastable lifetime $\tau$, which in turn depends on temperature $T$ and the field amplitude $H_0$. Several metastable states were also studied by P. A. Rikvold et al. [14] for an impurity-free kinetic Ising model with nearest neighbor interactions and local dynamics under a magnetic field in the framework of droplet theory and Monte Carlo simulation. They demonstrated that metastable lifetimes exhibited magnetic field and system-size dependencies. Additionally, W. D. Baez and T. Datta [13] established for a two-dimensional ferromagnetic kinetic Ising model with oscillating field and next-nearest neighbor interactions that metastable lifetimes are determined by the lattice size, amplitude and frequency of the external field, temperature and additional interactions present in the system.

Micromagnetic models [15] based on the solution of the Landau–Lifshitz–Gilbert (LLG) equation [16–18] have also allowed a theoretical description of micro and nanoscale magnetization processes above the exchange length. Micromagnetism integrates classical and quantum mechanical effects, where the spin operators of the Heisenberg model are replaced by classical vectors and also account for exchange interaction. The main assumptions of the model are that the distribution of the magnetic moments is considered discrete throughout the volume of the magnetic system by means of a set of discretization cells and, at the same time, it is approximated by a density vector $\mathbf{M}(\mathbf{r}, t)$, which is continuous and differentiable with respect to both space $\mathbf{r}$ and time $t$, and it can be written in terms of a unit vector field $\mathbf{M}(\mathbf{r}, t) = M_s\mathbf{m}(\mathbf{r}, t)$ where $M_s$ is the saturation value of a constant norm.

Despite all the studies devoted to systems exhibiting dynamic phase transitions, little effort has been paid to the role played by the spatial orientation of the easy-axis of a single nanoparticle relative to the direction and sense of the external applied oscillatory field $\mathbf{H}$. It must be stressed that different orientations of the magnetic easy-axis must be interpreted as different ensamble microstates occurring during Brown rotation, so by fixing the magnetic easy-axis in each microstate considered, we can drive our attention to the Néel relaxation at each Brown rotation step. Thus, in the present work the magneic sample is a single-domain magnetite nanoparticle of diameter 32 nm with the presence of an external AMF. Such a value lies in the range where magnetite nanoparticles exhibit both single-domain size and ferrimagnetic behavior, just above the superparamagnetic limit [19]. Due to the size of the nanoparticle, the only anisotropy considered was the uniaxial magnetocrystalline anisotropy, whose source is the spin–orbit interaction. Additionally, the contributions to the energy density are exchange, demagnetization, Zeeman and uniaxial magnetocrystalline anisotropy. We obtained the numerical solutions of the LLG equation. Throughout, micromagnetic simulations were performed using the Ubermag [20] package based on the Object Oriented Micromagnetic Framework (OOMF) [21]. Particular attention was devoted to the influence of the orientation of the magnetic easy-axis $\mathbf{u}$ of the MNP on the DPT. We calculated the metastable lifetimes for different amplitudes $H_0$ and orientations of the magnetic easy-axis $\mathbf{u}$ as well as the interplay between these two parameters. Moreover, the hysteresis losses at selected values of orientations of the magnetic easy-axis $\mathbf{u}$ were performed to evaluate the SLP of the magnetic sample.

We can't make a quantitative comparison between LLG results with the Stoner-Wholfarth's model [22] because the Stoner–Wohlfarth's model is a very simplistic one where the magnetization does not vary within the magnetic sample. It is very different to our model where the magnetization inside the particle is not uniform and it can vary, not only with time, but from point to point since we are interested in the dynamic effects produced by a time-dependent external field and not in the static ones as the Stoner–Wholfarth's model does where the applied field does not depend on time. Thus, it must be stressed that the $\mathbf{M}$-$\mathbf{H}$ cycles shown in our work are dynamic hysteresis loops and not static. This is the

reason we used the micromagnetic approach where a transcendental differential equation (LLG) of a time-dependent magnetization is solved. Moreover, our Hamiltonian involves both an exchange interaction through the stiffness constant for the coupling among discretization cells, and also a demagnetizing term standing for the magnetostatic interaction, which do not appear in the Stoner–Wholfarth's model.

The remainder of this paper is organized as follows. In Section 2, we present an overview of the micromagnetic background of a MNP with a given easy-axis of magnetization, its geometrical and physical aspects and computational details. Numerical results are presented in Section 3 where a proposal of a dynamic phase diagram is presented, and in Section 4 we make a summary and a brief conclusion of the main results.

## 2. Theoretical Model and Computational Details

### 2.1. Micromagnetic Energy

According to the micromagnetic theory, the energy density $w(\mathbf{M})$ is composed by a series of contributions according to the properties of the magnetic material. In our micromagnetic system, consisting of a single-domain magnetic nanoparticle of magnetite, contributions to the energy density are exchange, demagnetization, Zeeman and uniaxial magnetocrystalline anisotropy [23]:

$$w(\mathbf{M}) = -\frac{A}{M_s^2}\mathbf{M} \cdot \nabla^2\mathbf{M} - \frac{1}{2}\mu_0\mathbf{M} \cdot \mathbf{H}_d - \mu_0\mathbf{M} \cdot \mathbf{H} - \frac{K_1}{M_s^2}(\mathbf{M} \cdot \mathbf{u})^2 \tag{1}$$

where $A$ is the stiffness constant accounting for an effective ferromagnetic interaction, $\mathbf{H}_d$ is the demagnetizing field caused by magnetostatic interaction, $\mu_0 = 1.2566 \times 10^6 \text{ NA}^{-2}\text{m}^{-1}$ is the permeability of free space, $\mathbf{H}$ is the magnetic field, $K$ is the anisotropy constant and $\mathbf{u}$ denotes the magnetic easy-axis unitary direction. In this work, the magnetic field is a sinusoidal wave applied along the **z**-direction of the form $\mathbf{H} = H(t)\hat{z}$ with $H(t) = H_0\sin[2\pi f(t - t_0)]$, amplitude $H_0$, frequency $f$ and initial displacement $t_0$. The competition between the different micromagnetic energy contributions on minimization determines the equilibrium distribution of magnetization [24]. At each point or discretization cell of the magnetic sample, an effective field $\mathbf{H}_{\text{eff}}$ is calculated by means of:

$$\mathbf{H}_{\text{eff}} = -\frac{1}{\mu_0}\frac{\partial E(\mathbf{M})}{\partial \mathbf{M}} \tag{2}$$

with

$$E(\mathbf{M}) = \int_V w(\mathbf{M})dV \tag{3}$$

being the energy of the system over the entire volume $V$ of the sample.

### 2.2. Landau–Lifshitz–Gilbert (LLG) Equation

The time evolution of the magnetization of a magnetic nanoparticle with radius $R$ and saturation magnetization $M_s$ is ruled by the Landau–Lifshitz equation with Gilbert damping, which reads as follows [16–18],

$$\frac{d\mathbf{M}}{d\mathbf{t}} = -\frac{\gamma}{1 + \alpha^2}(\mathbf{M} \times \mathbf{H}_{\text{eff}}) - \frac{\gamma}{1 + \alpha^2}\frac{\alpha}{M_s}\mathbf{M} \times (\mathbf{M} \times \mathbf{H}_{\text{eff}}) \tag{4}$$

where $\alpha$ denotes the dimensionless damping parameter and $\gamma = 2.2128 \times 10^5 \text{ mA}^{-1}\text{s}^{-1}$ is the gyromagnetic ratio. The LLG equation is strictly valid only at absolute zero temperature and the normalized magnetization vector is $\mathbf{m} = \mathbf{M}/M_s$. The system transfers energy and angular momentum from the movement of the magnetic moments to other degrees of freedom through the $\alpha$ parameter [25]. The first term of the LLG equation is associated with the gyromagnetic ratio that promotes a uniform precession of the magnetization around the effective field. The second term is a phenomenological damping parameter that accounts for the energy loss of the system before reaching the equilibrium state.

### 2.3. Metastable Lifetime and Specific Loss Power (SLP)

Ferromagnetic systems attached to a periodically oscillating magnetic field can exhibit order-disorder DPT. A significant aspect in DPT is the metastable lifetime. To study this DPT behavior, first the system must be fully magnetized with all the magnetic moments aligned along the direction of the maximum external magnetic field. Once the field is allowed to change in time, we analyze the conditions under which the magnetization becomes reversed and it passes through the zero value of magnetization. The elapsed time between these two magnetic states corresponds to the metastable lifetime and it is schematically shown in Figure 2. This is a measure of the time it takes for the system to escape from the metastable saturation state in the free-energy landscape.

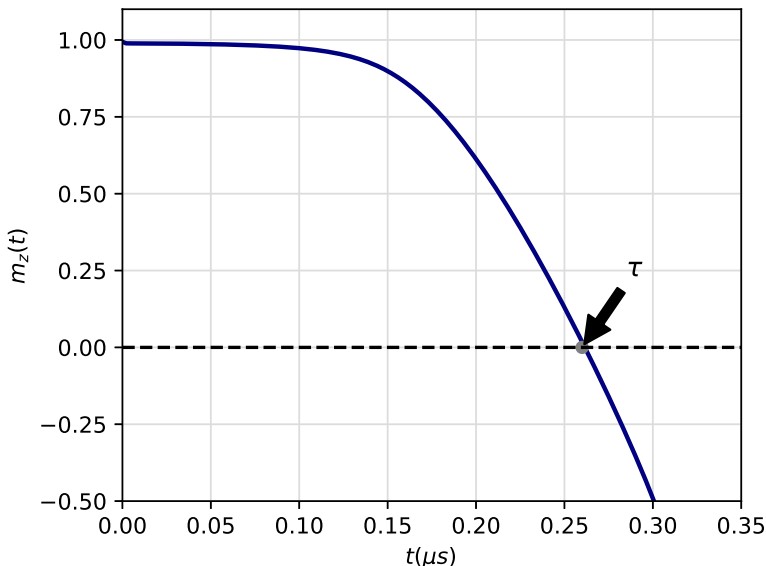

**Figure 2.** Determination of the metastable lifetime $\tau$ corresponding to the first-passage time through zero magnetization from saturation state. The initial configuration corresponds to a fully saturated state along the main direction of the AMF. The parameters are $f = 1$ MHz, $H_0 = 200$ mT and orientation of the magnetic easy-axis $\mathbf{u}_{82.5°}$ relative to the field direction. The metastable lifetime in this case is $\tau = 0.26$ μs.

A complete decay of the metastable phase occurs only in a dynamically disordered phase, as shown in Figure 1b. In this phase, the metastable lifetime is comparable with the period of the external magnetic field. The magnetization follows the magnetic field and its average over one period is close to zero. Consequently, the hysteresis loop is symmetric. The area enclosed by the hysteresis loop is the irreversible work dissipated in form of heat, which is quantified by the SLP energy. This heat is given by the following relationship [26,27]:

$$SLP = \frac{\mu_0 f}{\rho} \oint M(t) dH(t) \tag{5}$$

where $\rho$ is is the mass density of the particle. The integration is performed over one period of the oscillating magnetic field. The thermal energy quantified by the SLP is an important magnitude for magnetic hyperthermia studies [3]. The SLP depends on several parameters, such as $H_0$ and $f$ of the external field, composition and system size, shape and magnetic state of the MNP [8].

### 2.4. Computational Model

The simulation and the dynamic response of the magnetization of our micromagnetic system is carried out by means of the Ubermag micromagnetic software package [20,21].

The discrete representation of the magnetization distribution is obtained through the finite-difference method, as is schematically represented in Figure 3a, where we have considered a magnetic nanoparticle with pseudo-spherical geometry and a crystal structure corresponding to magnetite $Fe_3O_4$. The physical properties [28,29] used in our simulation for magnetite are shown in Table 1.

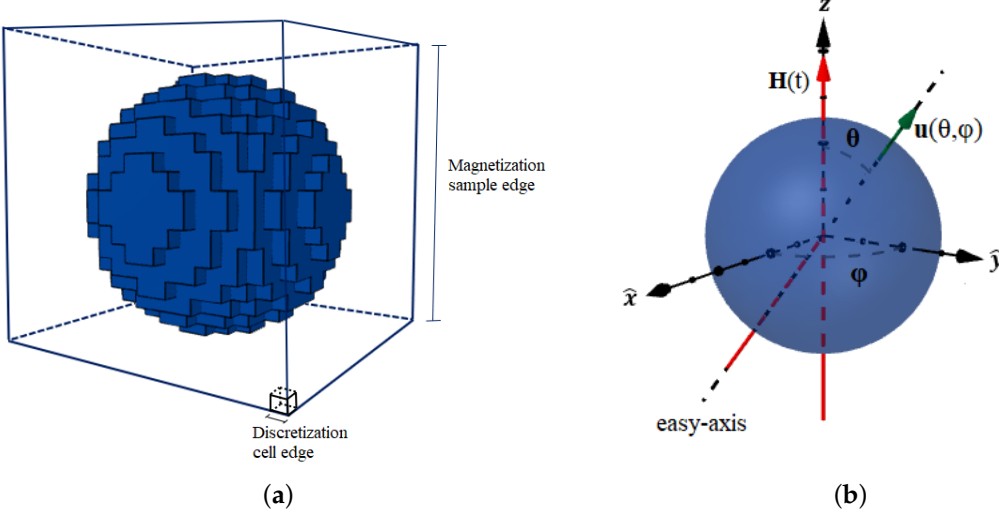

(a)                                                                                             (b)

**Figure 3.** (a) Finite-difference mesh used in our simulation. The magnetization sample edge is 36 nm and the discretization cubic cell edge is 2 nm. (b) Coordinates for the magnetic easy-axis orientation of the MNP. The MNP diameter is 32 nm, and $\theta$ is the angle between the main direction of the external AMF and the magnetic easy-axis orientation.

**Table 1.** Physical properties of magnetite.

| Chemical Formula | $M_s$ (Am$^{-1}$) | $A$ (Jm$^{-1}$) | $K$ (Jm$^{-3}$) | $l_{ex}$ (nm) | $\rho$ (Kgm$^{-3}$) |
|---|---|---|---|---|---|
| $Fe_3O_4$ | $4.46 \times 10^5$ | $7.00 \times 10^{-12}$ | $2.50 \times 10^4$ | 7.48 | 5240 |

The diameter used to simulate the magnetite nanoparticle was 32 nm. Value lies in the range where magnetite nanoparticles exhibit both single-domain size and ferrimagnetic behavior. Moreover, a time-dependent external field is applied. These features imply that the magnetization inside the particle is not uniform and it can therefore vary not only with time, but from point to point inside the particle with which a micromagnetic focus is valid. A discretization cubic cell of 2 nm edge was employed. This linear dimension is much smaller than the exchange length ($l_{ex}$) so an approach to the continuum is plausible. In the case of magnetite this is given in Table 1. A typical value of $\alpha = 0.07$ for a bulk ferrimagnetic material such as magnetite was employed [30]. The frequency of the AMF was $f = 1.0$ MHz, which is within the range of the radio-frequency currently used in magnetic hyperthermia [31]. Additionally, the initial displacement considered was $t_0 = -(4f)^{-1}$.

In this work, we consider only uniaxial anisotropy of the MNP. Moreover, initially the system is prepared in a fully positively magnetized state along the $\hat{z}$ direction and different orientations of the magnetic easy-axis **u** of uniaxial anisotropy are considered. In Cartesian coordinates, we have $\mathbf{u}(\theta, \varphi) = \sin(\theta)\cos(\varphi)\mathbf{x} + \sin(\theta)\sin(\varphi)\mathbf{y} + \cos(\theta)\mathbf{z}$, where the azimuthal angle was fixed at $\varphi = 90°$, and the polar angle $\theta$ range was $2.5° \leq \theta \leq 87.5°$.

## 3. Results and Discussion

The magnetization in the micromagnetic model is calculated by solving the LLG equation at zero temperature. First, we calculate the $z-$component of magnetization for amplitudes of $H_0 = 5$ mT and $H_0 = 200$ mT, as is shown in Figure 4a–c and Figure 4c–e, respectively. Simulations were performed for $\mathbf{u}_\theta = \mathbf{u}_{7.5°}$, $\mathbf{u}_\theta = \mathbf{u}_{45°}$ and $\mathbf{u}_\theta = \mathbf{u}_{82.5°}$. First,

note that for an amplitude of the external AMF of $H_0 = 5$ mT, and for each orientation of the easy-axis, the system is found in a DOP. In contrast, a DDP is evidenced for $H_0 = 200$ mT. Hence, the results shown in Figure 4 demonstrate that both DOP and DDP depend on the $H_0$ of AMF and the easy-axis orientation. A sharp reversal of magnetization takes place for $\theta$ values small, i.e., $\theta = 7.5°$.

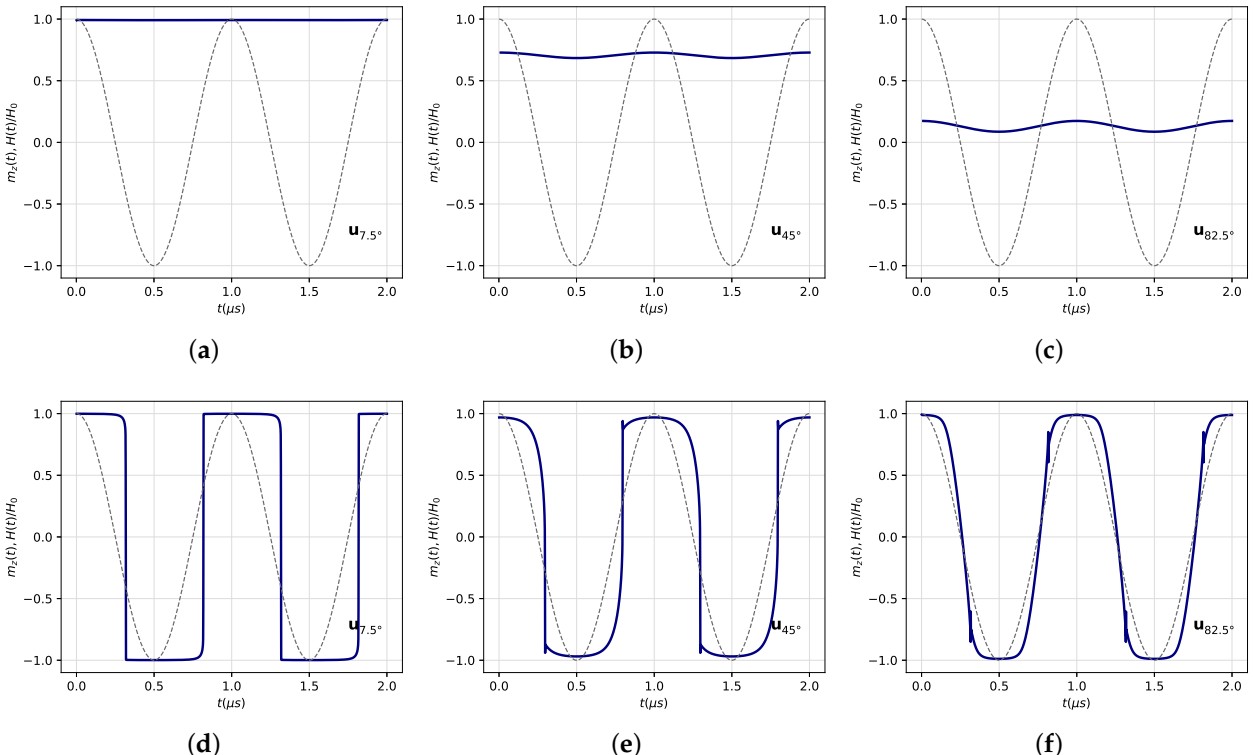

**Figure 4.** Time dependence of the $z$ component of magnetization and the external AMF. Polar angles for the easy-axis orientation are: (**a**) $\theta = 7.5°$, (**b**) $\theta = 45°$ and (**c**) $\theta = 82.5°$, for an amplitude of $H_0 = 5$ mT. Polar angles for the easy-axis orientation are: (**d**) $\theta = 7.5°$, (**e**) $\theta = 45°$ and (**f**) $\theta = 82.5°$, for an amplitude of $H_0 = 200$ mT. Figures (**a**–**c**) show a typical dynamic ordered phase (DOP) whereas figures (**d**–**f**) show a typical dynamic disordered phase (DDP).

Now that the dependence of DOP and DDP with the amplitude of the external AMF is known, we can calculate the critical amplitudes $H_{0C}$ for which the DPT of the type DOP-DDP occurs for a wide range of values of the polar angle $\theta$ $2.5° \leq \theta \leq 87.5°$, as shown in Figure 5. To achieve this dynamic magnetic phase diagram, it was necessary to systematically carry out several simulations such as those shown in Figure 4 in order to determine the critical amplitudes $H_{0C}$ of the AMF for which the magnetization was not able to be switched. As can be observed in this figure, the lower region corresponds to the DOP, whereas the upper region indicates the DDP. Thus, the DPT is strongly dependent on both the amplitude and the easy-axis orientation of the particle, and both quantities follow a non-linear relationship. Moreover, as the easy-axis is oriented nearly parallel to the magnetic field direction, the DPT takes place at higher values of amplitude. In contrast, for a nearly perpendicular orientation of $\mathbf{u}_\theta$, the DPT occurs at lower amplitude values. Thus, a nearly perpendicular orientation of $\mathbf{u}_\theta$ makes the magnetic system less rigid, giving rise to a DPT at small magnetic fields. This fact constitutes a way of tuning the magnetically hard or soft character of the system.

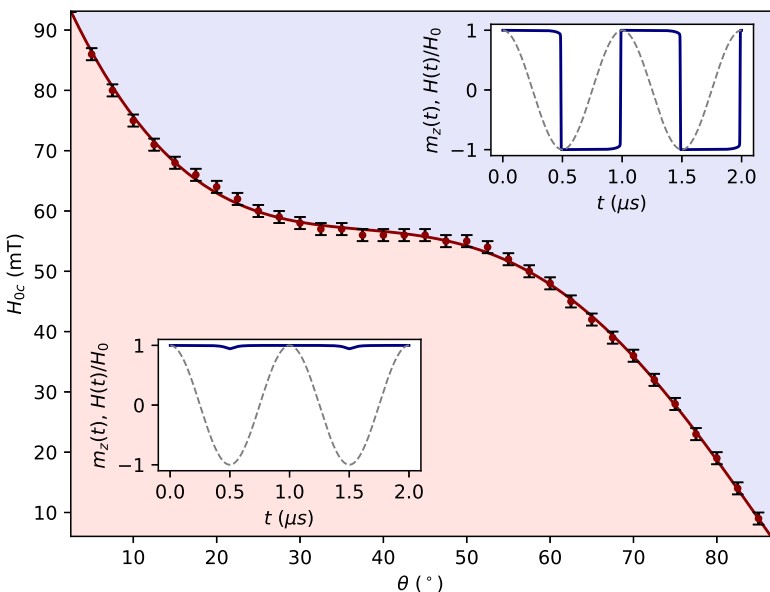

**Figure 5.** Dynamic phase diagram between the DOP and the DDP. The colored regions represent the DOP (pink) and DDP (violet) phases. Error bars are determined by the step of the AMF, namely $\Delta H_0 = 1$ mT.

As larger amplitudes of the external AMF are considered, the magnetization quickly escapes from its metastable state and $\tau$ slowly decreases as $\mathbf{u}_\theta$ is oriented nearly perpendicular to the external AMF direction. In contrast, the smaller the amplitude, the more time is required to escape from the metastable state, and $\tau$ quickly decreases with the increase in $\theta$. Thus, we conclude that the metastable lifetime $\tau$ not only depends on the amplitude of the external AMF at a given frequency, but also on the easy-axis orientation. Several numerical results for different orientations of $\mathbf{u}_\theta$ were obtained, as is shown in Figure 6.

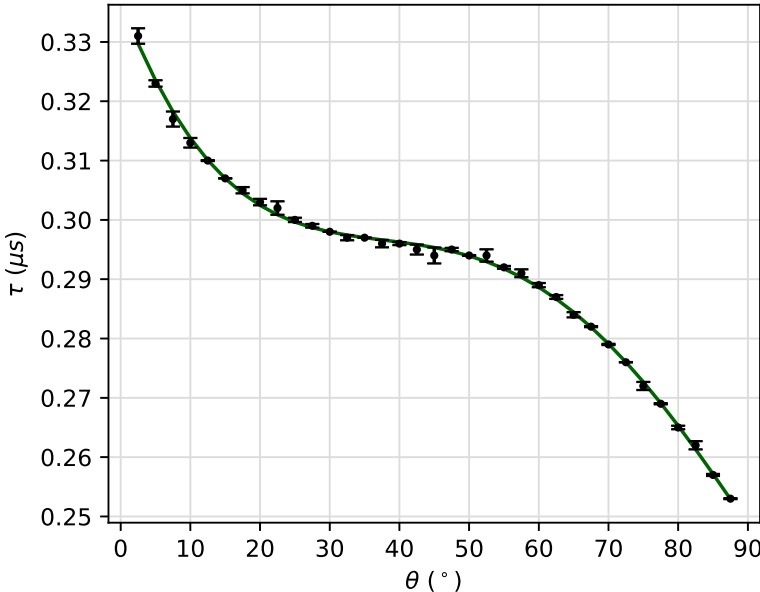

**Figure 6.** Dependence of the metastable lifetime $\tau$ with the polar angle $\theta$ for $H_0 = 200$ mT.

In addition, we also obtained the dynamic hysteresis loops at 0 K of temperature for different easy-axis orientations of at a given AMF field fixed amplitude of $H_0 = 200$ mT and fixed frequency of 1 MHz. The respective results are shown in Figure 7. Hence, as already unveiled, it is clear that the angle $\theta$ of the easy-axis strongly affects the magnetization

reversal. As long as the easy-axis is oriented nearly perpendicular to the external AMF main direction, coercivity is significantly reduced. This presumably implies that as the external field is applied in a direction away from that of the easy-axis, it is easier to "pull" the magnetization out of the easy direction of anisotropy through different paths in phase space. This fact is also responsible for the crossing of branches observed in the descending and ascending branches of the hysteresis loops at the switching fields, which is more noticeable for large values of $\theta$. This peculiar and surprisingly behavior of the crossing of hysteresis branches has been already demonstrated to occur in similar systems [32,33]. The crossing of the ascending and descending branches of the magnetization is observable when the applied external AMF is considerably far off the easy-axis, and this fact is associated with the energy minimum around the metastable state. Mathews et al. demonstrated in [32] that such a scenario can occur for a Stoner–Wohlfarth particle with a unique anisotropy. Hence, we have been able to show that the crossing of branches is significantly affected by the orientation of the particle.

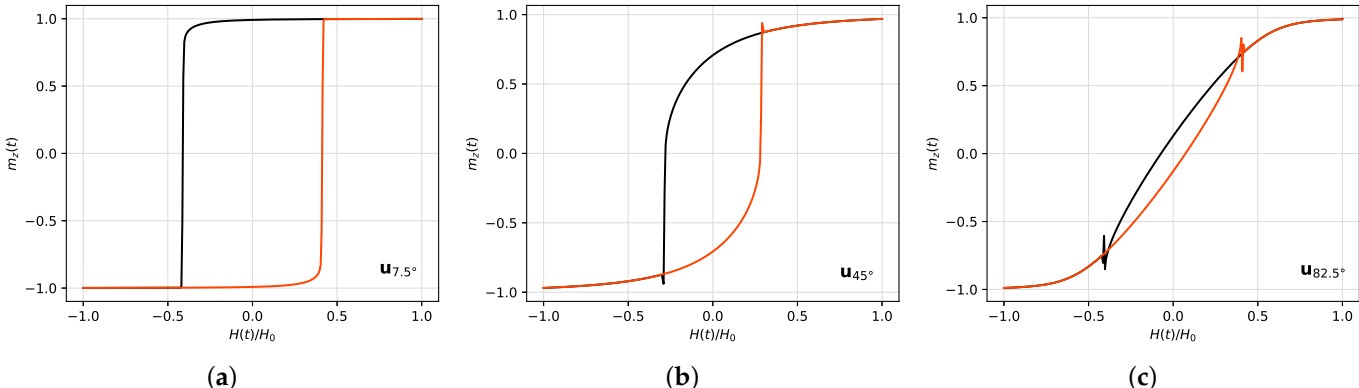

**Figure 7.** Hysteresis loops for a field amplitude $H_0 = 200$ mT and for three different angles of the applied field with respect to the easy-axis orientations of the MNP, namely (**a**) $\theta = 7.5°$, (**b**) $\theta = 45°$ and (**c**) $\theta = 82.5°$. The ascending curves (from negative to positive saturation) are shown in orange, while the descending curves (from positive to negative saturation) are shown in black. For $\theta = 45°$ and $\theta = 82.5°$, crossing of branches is evident.

On the other hand, the micromagnetic energy contributions for each orientation of the easy-axis are shown in Figure 8. Figure 8a–c shows the exchange energy for $\mathbf{u}_{7.5°}$, $\mathbf{u}_{45°}$ and $\mathbf{u}_{82.5°}$, respectively. Figure 8d–f stands for the demagnetization energy, Figure 8g–i for the uniaxial anisotropy energy and Figures 8j–l for the Zeeman energy and for the same angles. We observe that the main micromagnetic energy contribution is given by the Zeeman energy, whereas the exchange energy contribution is the smallest one.

For the case of $\mathbf{u}_{82.5°}$, a zoom of the uniaxial anisotropy and Zeeman energies is shown for clarity in Figure 9. Some jumps at $H(t)/H_0 = \pm 0.403$, $H(t)/H_0 = \pm 0.409$ and $H(t)/H_0 = \pm 0.414$ are observed for both the ascending and descending field branches, which is consistent with a sort of transition zone. In particular, a significant increase in the value of the uniaxial anisotropy energy is observed for $H(t)/H_0 = \pm 0.403$, which implies that the magnetic moments are not aligned with the easy-axis of the MNP. At the same value of $H(t)/H_0$, a minimum in the Zeeman energy is observed, which implies an alignment of the magnetic moments with the field direction. In contrast, for $H(t)/H_0 = \pm 0.409$, the opposite scenario takes place. This means that during the transition zone, magnetic moments become magnetically frustrated, as a consequence of the competition between these two energies.

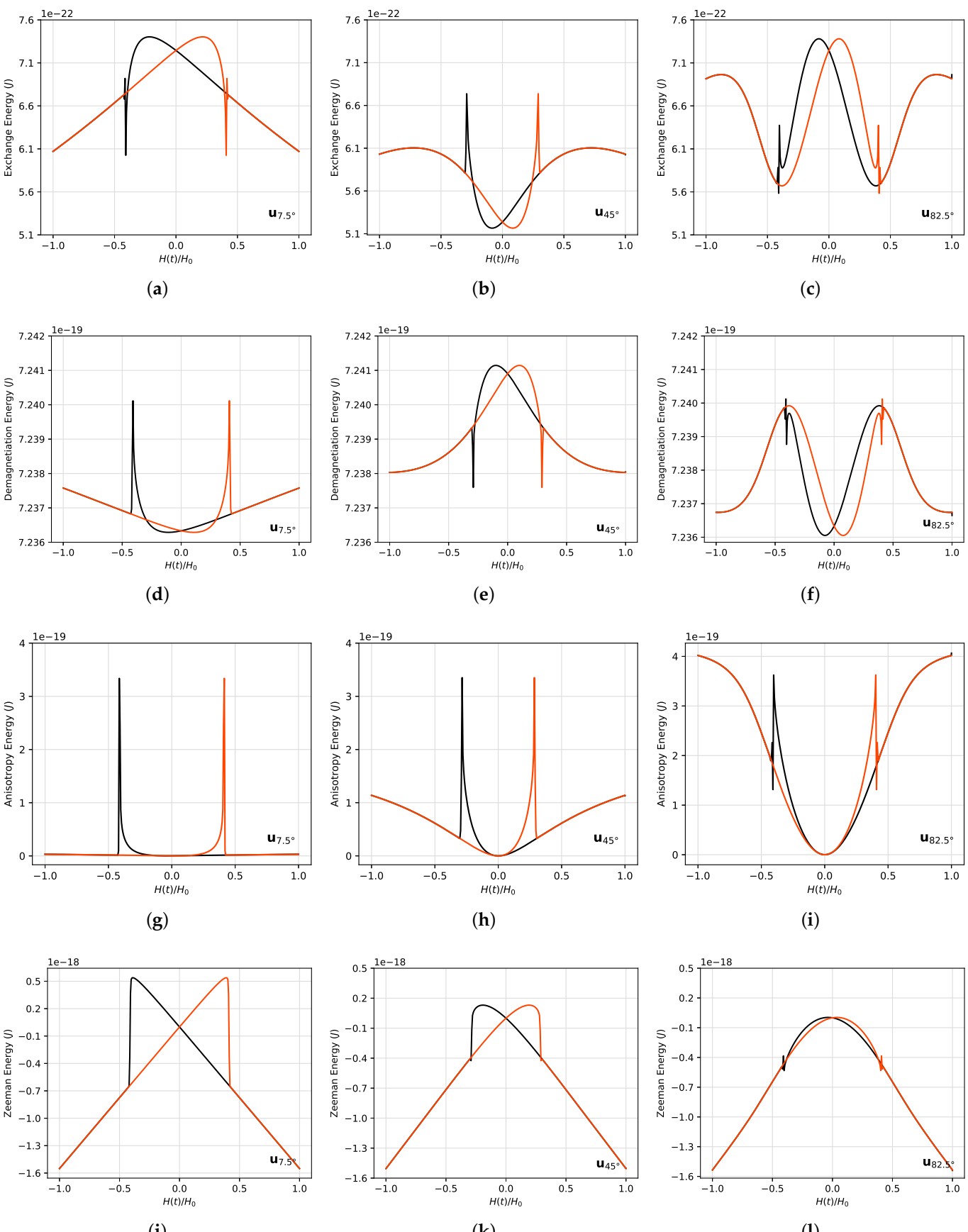

**Figure 8.** Micromagnetic energy contributions as a function of the reduced applied field for different easy-axis orientations $\mathbf{u}_{7.5°}$, $\mathbf{u}_{45°}$ and $\mathbf{u}_{82.5°}$. (**a–c**) Exchange energy. (**d–f**) Demagnetization energy.

(**g–i**) Uniaxial anisotropy energy. (**j–l**) Zeeman energy. Orange lines stand for the ascending curves, while the descending curves are shown in black lines.

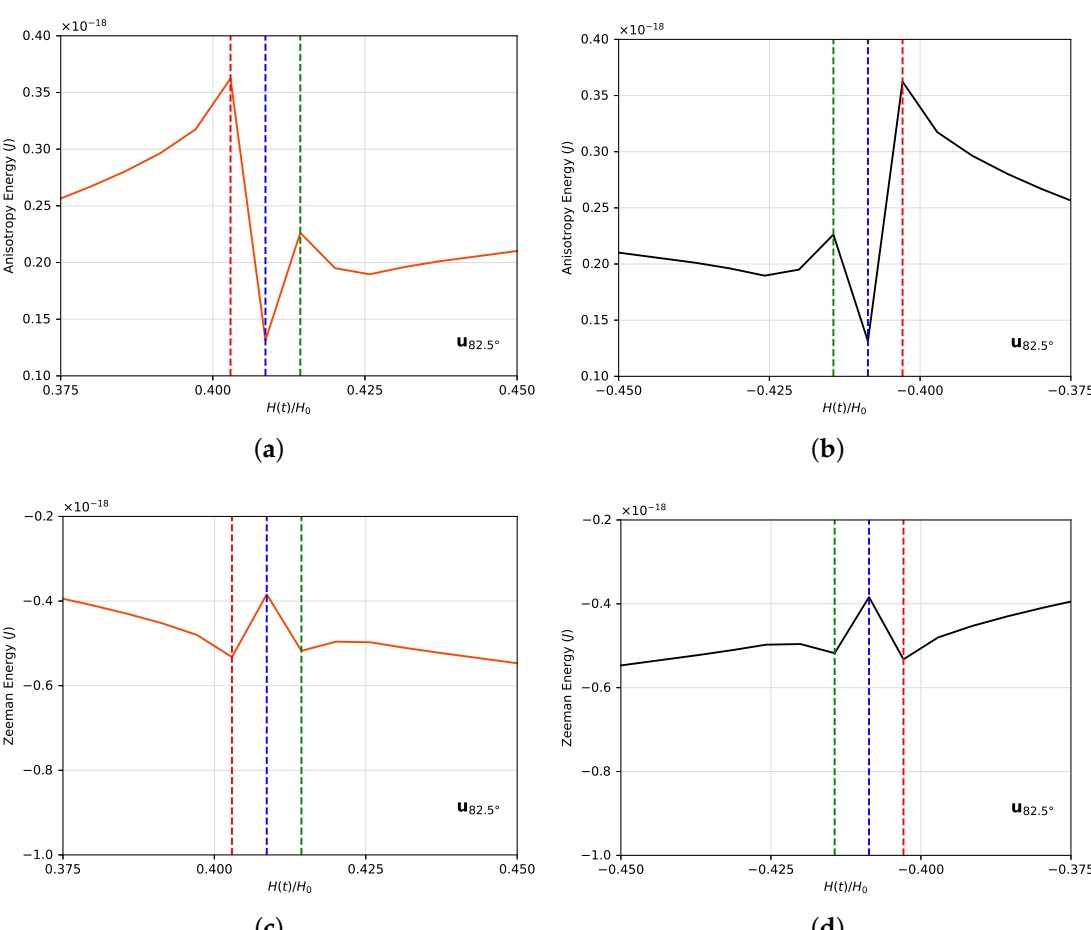

(**a**)          (**b**)

(**c**)          (**d**)

**Figure 9.** Zoom of the (**a**,**b**) uniaxial anisotropy and (**c**,**d**) Zeeman energies as a function of $H(t)/H_0$ for $\mathbf{u}_{82.5°}$. The ascending field branch is depicted in orange whereas the descending field branch is shown in black. At $H(t)/H_0 = \pm 0.403$ (red dashed line), $H(t)/H_0 = \pm 0.409$ (blue dashed line) and $H(t)/H_0 = \pm 0.414$ (green dashed line), energy contributions exhibit some jumps.

Finally, all of the above results lead to ask about the amount of heat that can be dissipated in terms of the specific loss power. Accordingly, the area enclosed by hysteresis loops was calculated to estimate the SLP by means of the hysteresis losses. Such quantity was computed using Equation (5), and the respective angular dependence is shown in Figure 10. The SLP is maximized when the easy-axis of the MNP is oriented nearly parallel to the magnetic field direction, that is, for small angle values there is a greater efficiency in the amount of heat released to the environment. This is a novel result because in the literature it is known that the specific loss power depends on parameters such as the size of the particles, the frequency and the amplitude of the external AMF [8,9], but to the best of our knowledge, no studies of this specific regard have been addressed up to now. This result may shed light on new ways to maximize the SLP for hyperthermia purposes.

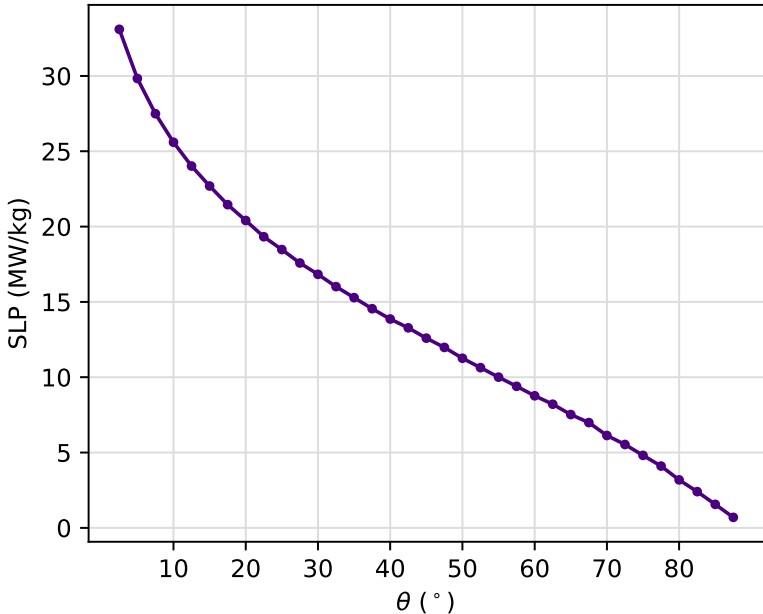

**Figure 10.** Variation of the SLP as a function of the angle $\theta$ of easy-axis orientation for an amplitude $H_0 = 200$ mT of the external AMF.

## 4. Conclusions

In this study, we have presented numerical calculations of the micromagnetic properties and dynamic critical behavior of the magnetization for a MNP of magnetite under the influence of an alternating magnetic field. The diameter used to simulate the magnetite nanoparticle of 32 nm lies in the range where magnetite nanoparticles exhibit both single-domain size and ferrimagnetic behavior. Moreover, a time-dependent external field is applied. Exchange, demagnetization, uniaxial anisotropy and Zeeman energies were considered in the Hamiltonian. The exchange length is much smaller compared with the size of the particle so an approach to the continuum is possible. Thus, the magnetization inside the particle is not uniform and it can therefore vary, not only with time, but from point to point inside the nanoparticle. The numerical solution of the LLG equation at zero temperature provides information on how the magnetization evolves in time and dynamic hysteresis loops can be obtained. The numerical simulation was performed by means of the micromagnetic simulator, the Ubermag package, based on the Object Oriented Micromagnetic Framework (OOMMF). We studied the effect of the orientation of the easy-axis of the MNP upon the critical values of the amplitude of the external AMF dealing with a dynamic phase transition. Thus, results were summarized in a proposal of a dynamic magnetic phase diagram. In addition, we calculated the time it takes for the magnetization to achieve the first-passage through the zero magnetization value from a saturated state, called the metastable lifetime, for different angles between the easy-axis and the direction of the external applied field. For a fixed amplitude $H_0 = 200$ mT, a fixed frequency $f = 1$ MHz and a range of polar angle $2.5° \leq \theta \leq 87.5°$, we obtained the hysteresis loops and calculated the specific loss power, which is proportional to the hysteresis loop area, to know the amount of heat that can be potentially released to the medium. Our results show that the dynamic phase transition, the metastable lifetime and the specific loss power, strongly depend on the amplitude of the external field and the orientation of the particle. Furthermore, our results show that the crossing of hysteresis branches are also affected by such orientation. Finally, SLP results may shed light on new ways to maximize this quantity for hyperthermia purposes by tuning the orientation of the particles in the system.

**Author Contributions:** Conceptualization, J.R. and N.R.; methodology, N.R.; software, N.R.; validation, J.R. and N.R.; formal analysis, N.R.; investigation, J.R. and N.R; writing—original draft preparation, N.R.; writing—review and editing, J.R.; supervision, J.R.; project administration, J.R.; funding acquisition, J.R. All authors have read and agreed to the published version of the manuscript.

**Funding:** This research was funded by Universidad de Antioquia through the CODI-Projects with grant numbers 2020-34211, 2022-51311, 2022-51312 and 2022-51330.

**Data Availability Statement:** Not applicable.

**Acknowledgments:** Financial support was provided by the CODI-UdeA Projects 2020-34211, 2022-51311, 2022-51312 and 2022-51330. One of the authors (J.R.) acknowledges the University of Antioquia for the exclusive dedication program.

**Conflicts of Interest:** The authors declare no conflict of interest.

## Abbreviations

The following abbreviations are used in this manuscript:

| | |
|---|---|
| MNP | Magnetic nanoparticles |
| AMF | Alternating magnetic field |
| DPT | Dynamic phase transition |
| SLP | Specific loss power |
| DOP | Dynamically ordered phase |
| DDP | Dynamically disordered phase |

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
