# Peer review of "Micromagnetic Approach to the Metastability of a Magnetite Nanoparticle and Specific Loss Power as Function of the Easy-Axis Orientation"

_2673-7167, doi:10.3390/physchem3030020_

Round 1

Reviewer 1 Report

This manuscript presents micromagnetic simulations of magnetic nanoparticles in alternating fields, through solutions of the Landau-Lifshitz-Gilbert equation. While results are sound, and also well-known from much simpler simulations based on the Stoner-Wohlfarth model, there are some points that need clarification:

1) Magnetic parameters for magnetite (Table 1) are used to simulate spherical particles (please mention the diameter, apparently 25 nm from Fig. 3a). Spherical particles do not have a shape anisotropy, but the anisotropy given in Table 1 must is obviously uniaxial, considering the solutions obtained. What is the source of anisotropy in the particles? If just arbitrarily assumed, then it must be mentioned, along with the fact that magnetite's magnetocrystalline anisotropy has not been considered.

2) A 25 nm maprticle with the properties of Table 1 reverses its moment by uniform rotation and is therefore well represented by thed Stoner-Wohlfarth model. Indeed, results are very similar, if not identical, to those that can be obtained with the much simpler Stoner-Wohlfarth model assuming a uniform particle magnetization. Is the use of the LLG equation justified? How do the LLG results compare quantitatively with the Stoner-Wohlfarth model?

Reviewer 2 Report

The publication of Roa and Restrepo use the micromagnetic approach to underline the effect of easy-eay-axis orientation on the SLP.

Before publication some minor issue can be clarified:

1/ Authors can justify in which condition micromagnetic approach can be use to nanoparticles

2/ The authors can explain the assumptions for the calculations (single-domain nanoparticles, coherent spin rotation, ferromagnetic or superparamagnetic nanoparticles).

3/ Can the authors provide a reference to justify the data presented in table 1?

There are a few typographical errors in the manuscript.
